



# A Simple and Realistic Aerosol Emission Approach for use in the Thompson-Eidhammer microphysics scheme in the NOAA UFS Weather Model (version GSL global-24Feb2022)

Haiqin Li[1,2], Georg. A. Grell[2], Ravan Ahmadov[2], Li Zhang,[1,2] Shan Sun[2], Jordan Schnell[1,2] and Ning Wang[2,3]

[1]Cooperative Institute for Research in Environmental Sciences at the University of Colorado Boulder, USA
[2]NOAA/Global Systems Laboratory/Earth Prediction Advancement Division, USA
[3]Cooperative Institute for Research in the Atmosphere at the Colorado State University, USA

*Correspondence to*: Haiqin Li (Haiqin.Li@noaa.gov)

**Abstract.** A physics suite under development at NOAA's Global System Laboratory (GSL) includes the aerosol-aware
double moment Thompson-Eidhammer microphysics scheme (TH-E MP). This microphysics scheme uses two aerosol variables (water friendly (WFA) and ice friendly (IFA) aerosol number concentrations) to include interaction with some of the physical processes. In the original implementation, WFA and IFA depend on emissions derived from climatologies. In our approach, using the Common Community Physics Package (CCPP), we embedded sea-salt, dust, and biomass burning emission modules as well as anthropogenic aerosol emissions into the Unified Forecast System (UFS) to provide realistic
aerosol emissions for these two variables. This represents a very simple approach with no additional tracer variables and therefore very limited additional computing cost. We then evaluate a comparison of simulations using the original TH-E MP approach, which derives the two aerosol variables using empirical emission formulas from climatologies (CTL) and simulations that use the online emissions (EXP). Aerosol Optical Depth (AOD) is derived from the 2 variables and appears quite realistic in the runs with online emissions when compared to analyzed fields. We find less resolved precipitation over
Europe and North America from the EXP run, which represents an improvement compared to observations. Also interesting are moderately increased aerosol concentrations over Southern Ocean from the EXP run invigorating the development of cloud water and enhances the resolved precipitation in those areas. This study shows that a more realistic representation of aerosol emission may be useful when using double moment microphysics schemes.

## 1 Introduction

The abundant aerosols in the atmosphere are able to influence both weather and climate through cloud formation and precipitation. The change of cloud drops size and cloud drop number have impacts on cloud albedo and energy budget (Twomey 1977), the hydrometeors content, cloud cover and lifetime (Albrecht 1989), and they can further suppress or enhance precipitation (Baklanov et al. 2014). Aerosol interaction with clouds – because of its impact on radiation termed the
indirect effect - still represents a large uncertainty for global climate forcing. Although there have been literally hundreds of papers looking into the effect on cloud and precipitation, the science is still not settled for sure, and still need to be investigated (Tao et al. 2012).





In recent decades, numerous studies have been conducted to research this effect with numerical models. Hoffmann and Feingold (2021) coupled a Lagrangian cloud model to a parcel model and a large-eddy simulation model to study the marine stratocumulus. They found that the selection of seeded particle size distribution is essential to the success simulation of marine cloud brightening, and its efficacy is significantly affected by microphysical processes. Conrick et al. 2021 evaluated the influences of wildfire smoke and cloud microphysics during a Pacific Northwest Wildfire case. They found that the thermodynamic changes due to smoke are the primary driver of enhanced cloud lifetime during the wildfire events, even more so than the microphysical impacts to clouds, which is the secondary contributing factor, and also pointed out that both the thermodynamic and microphysical effects are necessary. Kang et al. (2019) coupled a simplified chemistry package into the Global/Regional Integrated Model system (GRIMs, Hong et al. 2013), and converted the soluble chemical species to the cloud condensation nuclei (CCN) for the cloud precipitation physics. When coupling with the simplified chemistry package, the cloud water increases and results in a decrease of surface downward shortwave radiation, but the precipitation responses to aerosol is not monotonic. Zhao et al. 2021 introduced the marine organic aerosols (MOA) as a new aerosol into the Community Atmosphere Model version 6 (CAM6), and also implemented the MOA for droplet activation and ice nucleation. In their study the marine ice nucleating particles are dominant below 400 hPa over the Southern Ocean and Arctic boundary layer. Over the Southern Ocean, the shortwave cloud forcing is reduced in the austral summer, and the longwave cloud forcing is enhanced in the austral winter when MOA acts as CCN and ice nucleating particles respectively. Mulcahy et al. 2014 evaluated a hierarchy of aerosol representations of aerosol climatology, fully prognostic aerosols, and initialized aerosols with data assimilation within the UK Met Office Unified Model (MetUM). They found that aerosol impacts on global precipitation and large-scale circulation appear small in the short-range forecasts, yet indirect aerosol effect can have significant impacts in local regions. They highlight the importance to include a realistic treatment for aerosol-cloud interactions in global short-range forecast models and the possibility to improve predictions by incorporating aerosol schemes. Grell et al. (2011) used a double moment microphysics scheme with significantly more complex chemistry and aerosols in WRF-Chem (Grell et al. 2005). For stable precipitation areas, coverage and intensity of precipitation were depressed, but the cloud-water mixing ratio and number concentrations are enhanced by the wildfire smoke in their study. Grell et al. (2011) showed that for deep convection, the behavior of the response was very different, sometimes even opposite. This study will focus mostly on non-convective precipitation in the higher latitudes.

At NOAA Global Systems Laboratory (GSL), in collaboration with the NOAA Chemical Science Laboratory (CSL) and Air Resource Laboratory (ARL), an atmospheric composition suite (based on WRF-Chem) was developed and coupled with the UFS Weather Model through the National Unified Operational Prediction Capability (NUOPC)-based NOAA Environmental Modeling System (NEMS) software. This modeling system has been operational as an ensemble member of the Global Ensemble Forecast System (named as GEFS-aerosols, Zhang et al. 2022) for global aerosol predictions. When using the NUOPC coupler, there are two independent components for atmosphere and chemistry that communicate via the coupler every time-step. Because of the interactive and strongly coupled nature of chemistry and physics, it is natural to allow for some of the atmospheric composition modules to be called directly from inside the physics suite. This can be



accomplished through the use of the Common Community Physics Package (CCPP, Heinzeller et al. 2023), which is designed to facilitate a host-model agnostic implementation of physics parameterizations and has been used by many different organizations. All the physics parameterizations in the Unified Forecast System (UFS) Weather Model are CCPP-compliant. In this study the low-level chemical routines were embedded directly into UFS Weather Model using CCPP (Li et al. 2021). A physics suite is under development at GSL, which includes the aerosol-aware double moment Thompson-Eidhammer microphysics scheme (TH-E MP, Thompson and Eidhammer 2014), and the scale-aware and aerosol-aware Grell-Freitas convection scheme (GF, Grell and Freitas 2014; Freitas et al. 2021). This study represents an examination of the applicability of a very simple approach to represent the aerosol indirect effect in a global modeling system as originally developed by TH-E. However, instead of using the aerosol climatologies that TH-E embedded in their MP scheme, to derive the two necessary aerosol variables in their scheme, we evaluate the differences when implementing realistic aerosol emissions to fill the two variables.

This paper is organized as follows. The model, method and experiment design are described in section 2. The results are presented in section 3, followed by a discussion section. Finally, the summary and future plans are given in section 5.

## 2 Model and Experimental Design

### 2.1 The UFS Weather Forecast Model

The Unified Forecast System (UFS, Jacobs 2021) Weather Model is a short- and medium-range research and operational forecast model that may be used across global and regional scales. It employs the Finite-Volume Cubed-Sphere (FV3, Lin 2004). The software infrastructure Flexible Modeling System (FMS) is used for functions such as parallelization. The CCPP is used for physical parameterizations and to connect them to the host model. The main program is created by the NOAA Environmental Modeling System (NEMS) model driver. The UFS Weather Model (https://github.com/NOAA-GSL/ufs-weather-model/releases/tag/global-24Feb2022) used in this study is a fork from the authoritative repository (https://github.com/ufs-community/ufs-weather-model). Thus, the version/release of GSL global-24Feb2022 does not come from the authoritative repository but from a fork at NOAA GSL.

### 2.2 Physics and aerosol emission

The microphysics used in this study is the aerosol-aware Thompson-Eidhammer Microphysics (TH-E MP, Thompson and Eidhammer 2014) scheme. It includes prognostic fields of mixing ratios of the hydrometeors of cloud water (Qc), cloud ice (Qi), rain water (Qr), snow water (Qs) and graupel (Qg), and the number concentrations of prognostic cloud water, cloud ice, and rain water. In TH-E MP, the hygroscopic aerosol is referred as a "water friendly" aerosol (WFA), and the non-hygroscopic ice-nucleating aerosol is referred as "ice friendly" aerosol (IFA). A Semi-Lagrangian sedimentation is implemented into TH-E MP to replace the Eulerian sedimentation to allow for larger time-steps in global modeling (Hong et al. 2022).

In our study, the sea-salt, dust emission, biomass burning emission modules, and the anthropogenic aerosol emissions are embedded into the UFS Weather Model as physics subroutines. Dust emissions are from the FENGSHA dust





scheme (Tong et al. 2017), and the sea salt emissions are from the NASA GOCART model. The anthropogenic organic carbon emissions are from the emission inventories of the Community Emissions Data System (CEDS, Hoesly et al. 2018). Fire Radiative Power (FRP) from the blended Global Biomass Burning Emissions Product (GBBEPx, Zhang et al. 2014) is used to provide biomass burning emissions. All modules are CCPP-compliant and are called directly within the physics

block, before application of the boundary layer and convection parameterizations. The WFA tendency and IFA tendency are calculated following Ackermann et al. (1998) and Powers et al. (2017) as in Equations 1a and 1b, respectively, from the sea-salt, organic carbon, and dust emissions.

$$\frac{dWFA}{dt} = \left[\frac{emis\_ss}{\rho_{ss}} * fact_{wfa\_ss} + \frac{emis\_oc}{\rho_{oc}} * fact_{wfa\_oc}\right] * \rho_{sfc} * dz_{sfc} \qquad \text{(Eq. 1a)}$$


$$\frac{dIFA}{dt} = \left[\frac{emis\_dust}{\rho_{dust}}\right] * fact_{ifa} * \rho_{sfc} * dz_{sfc} \qquad \text{(Eq. 1b)}$$

$$fact_{wfa\_ss} = (1^{-9} * \frac{6}{pi})^{4.5*log1.8^2}/(wfa\_diameter\_ss)^3 \qquad \text{(Eq. 2a)}$$

$$fact_{wfa\_oc} = (1^{-9} * \frac{6}{pi})^{4.5*log1.8^2}/(wfa\_diameter\_oc)^3 \qquad \text{(Eq. 2b)}$$

$$fact_{ifa} = (1^{-9} * \frac{6}{pi})^{4.5*log1.8^2}/(ifa\_diameter)^3 \qquad \text{(Eq. 2c)}$$

where $emis\_ss$ is the sea salt emission, and $emis\_oc$ is the combined organic carbon from anthropogenic emission and
wildfire. The dust emission is represented by $emis\_dust$. $\rho_{sfc}$ is the model surface layer air density, and $dz_{sfc}$ is the model surface layer depth. $\rho_{ss}$ , $\rho_{oc}$ and $\rho_{dust}$ are density of sea salt, organic carbon, and dust respectively.

The GF cumulus convection scheme (Grell and Freitas 2014, Freitas et al. 2021) is used to treat non-resolved convection in this study. The GF is a scale-aware, can be used as an aerosol-aware convective scheme, and allows for shallow, congestus and deep convective modes. The convective wet scavenging and convective transport of WFA and IFA
are included in the GF scheme. In order to isolate the microphysics response to aerosols, the aerosol-aware feature of GF is turned off in this study. The Mellor-Yamada-Nakanishi-Niino (Nakanishi and Niino 2009) eddy-diffusivity mass-flux (MYNN-EDMF, Olson et al. 2019) planetary boulder scheme is used to represent PBL transport.

**2.3 Numerical Experiment design**

There are two sets of modeling experiments in this study. One is the control run (CTL hereafter), which uses the
default aerosol emission and GOCART WFA/IFA initial conditions for the TH-E MP. An empirical WFA emission power formula, which depends on the surface number concentration (Thompson and Eidhammer 2014), is used to calculate the WFA tendency, and there is no IFA emission in the CTL run. A second experiment (EXP hereafter) uses WFA and IFA initial conditions converted from the operational GEFS-Aerosol. This provides a better set of real-time initial conditions since in addition to sea salt, dust, and carbon, sulfate is also available for the EXP experiment in the initial conditions. The



WFA tendency and IFA tendency are calculated online as described in section 2.2. Then the prognostic WFA and IFA tendencies are added to the surface layer WFA and IFA every time step.

     Both runs use atmospheric initial conditions from the GFS analysis. There are ten boreal winter runs initialized every 120 h from 00 UTC of December 1st 2020 to 00 UTC of January 15th 2021, and the forecast is integrated for 120 h. We selected the boreal winter period to minimize the impact of convective precipitation over the Northern Hemisphere, and to

focus on the sensitivity of the resolved precipitation to aerosols. The horizontal resolution is C768 (~13km) and there are 127 vertical layers. The GFS Near-Surface Sea Temperature (NSST) scheme is used to provide Sea Surface Temperature (SST) forecast during the model integration.

**2.4 Datasets for Forecast Verification**

     Two aerosol reanalysis products from the European Center for Medium Range Weather Forecasts - Copernicus

Atmosphere Monitoring Service (ECMWF-CAMS, Inness et al. 2019), and the Modern-Era Retrospective analyses for Research and Applications, version 2 (MERRA2, Ronald et al. 2017) are used to validate the Aerosol Optical Depth (AOD). The radiation flux is validated by the Clouds and the Earth's Radiant Energy Systems (CERES, Rutan et al. 2015) observation dataset. The NOAA Climate Prediction Center (CPC) Global Unified Gauge-Based Analysis of Daily Precipitation (Chen and Xie, 2008) is used to verification the precipitation forecast. The GFS analysis is also used to

calculate the 500 hPa geopotential height Anomaly Correlation Coefficients (ACC).

**3 Results**

     We are presenting retrospective run results from on average over 10-cases. The WFA, IFA, AOD, temperature, and hydrometeors are averaged at 120 h into the forecast. The cloud cover, radiation, and precipitation are from the average values over the 120-h forecast period.

**3.1 Water-Friendly aerosols and Ice-Friendly aerosols**

     The average of the 120-h forecast of WFA from the CTL runs (Fig. 1a) shows the largest concentrations mainly over the oceanic areas of equatorial Atlantic and northern Indian Ocean, and over the land areas of the Amazon, central Africa, and South and East Asia. WFA from the EXP runs (Fig. 1b) appears larger over land than over ocean. For example, we can observe high WFA over eastern North America, Central America, Euro-Asian continent and Savanna and Rainforest

Africa, with the difference between EXP and CTL (Fig. 1c) indicating that the EXP runs have less WFA especially over the tropical oceans, and in general more WFA over the Northern Hemisphere as well as slightly more over the high latitude Southern Oceans. The maximum of IFA from the CTL runs (Fig. 2a) is located over tropical Atlantic, but significant concentrations cover most aeras of Northern Hemisphere. However, high IFA concentrations from the EXP runs (Fig. 2b) concentrates mostly over the desert areas of the Sahara, with some dust emissions leading to elevated IFA concentrations

over Australia and Southern South America, and Eastern Asia.

**3.2 Aerosol Optical Depth**

     Aerosol optical depth (AOD) is a key variable to measure the absorbing and scattering radiation by aerosol particles. In this study analyzed 550nm AOD fields are used to compare with simulated 550nm AOD fields derived from





WFA and IFA by using a look-up table (Colarco et al., 2010) with observation constraint. Two kinds of reanalysis data,
MERRA2 (Ronald et al. 2017, Fig. 3a) and ECMWF-CAMS (Inness et al. 2019, Fig. 3b), are used in this study to evaluate
the average of 120-h forecast AOD from the CTL runs (Fig. 3c) and the EXP runs (Fig. 3d). The AOD from the CTL runs is
high over tropical Atlantic and northern India Ocean. This overestimation is consistent with the distribution of WFA and
IFA, and the CTL runs also underestimate the AOD over South Asia, East Asia, and mid-latitude ocean areas (Table 1). The
absence of IFA emission in the CTL runs may result in weaker IFA and AOD over the Sahara area. The EXP runs (Fig. 3d)
capture the AOD over continental areas of central Africa, South Asia and East Asia as well as the observed AOD distribution
over the northern Pacific, northern Atlantic and the Southern Ocean. However, AOD is overestimated over Europe and the
Eastern part of the US. Large uncertainties exist in the two analysis fields, with MERRA2 analysis probably in closest
agreement, indicating higher AOD compared to the CAMS analysis over Europe (Table 1). Overall, the AOD derived from
WFA and IFA is in reasonable agreement with analysis, suggesting it may be worth to also investigate the direct radiation
impact in comparison to using climatologies.

### 3.3 Cloud, radiation and liquid water path

Most areas of the Earth are covered by clouds, and aerosol plays a key role in the formation of clouds. The 120-h
averaged high cloud, mid cloud and low cloud cover fractions (Yoo and Li, 2012) from the CTL runs are shown in Figure
4a, 4b and 4c respectively. Compared with the CTL runs, the EXP runs have more high cloud cover over most continental
areas of North America and Euro-Asia (Fig. 4d), more mid-cloud cover over Northern Hemisphere and less over tropical
aeras (Fig. 4e), more low cloud cover over the mid-latitude areas of Northern Hemisphere (Fig. 4f). This appears physically
consistent with the WFA differences in Figure 1c. However, there are no significant low cloud cover differences over high-
latitude aeras, and this is probably because the model forecasts both have low cloud cover close to 100% over high-latitude
aeras (Fig. 4c).

The 120-h averaged outgoing longwave radiation (OLR) bias from CTL against CERES observation is shown in
Figure 5a. In general, CTL has positive OLR bias over land, and negative bias over tropical ocean areas. The EXP runs
slightly improve the tropics bias (Fig. 5b), e.g., less OLR over tropical land and more OLR over tropical ocean. The average
global mean OLR from the CERES observation in comparison to the CTL and EXP experiments are 234.5 W m$^{-2}$, 237.0
W/m$^2$, and 237.1 W m$^{-2}$ respectively. The 120-h averaged surface downward shortwave radiation (SFCDSW) biases from
the CTL runs relative to CERES observation is shown in Fig. 6a, and Fig. 6b shows the SFCDSW differences between the
EXP and CTL runs. While it may be more difficult to see clear improvements in the EXP runs, there are interesting and
notable differences. The EXP runs have less SFCDSW over the northern hemisphere and more SFCDSW over tropical areas,
consistent with the cloud fraction differences. There are less SFCDSW from the EXP runs over the Southern Ocean,
although there are no significant cloud fraction differences between the EXP and CTL runs. The global mean SFCDSW from
the averaged CERES observations compared to the CTL and EXP runs are 192.5 W m$^{-2}$, 194.7 W m$^{-2}$ and 193.7 W m$^{-2}$
respectively.



Liquid water path (LWP) is important for cloud radiation and precipitating processes (Gryspeerdt et al. 2019). The LWP from the CTL run is shown in Figure 7a, with the largest LWP over the northern and western Pacific, northern Atlantic, Europe and the Southern Ocean. The EXP run generates more LWP than the CTL run over northern Pacific, northern Atlantic, and Southern Ocean, and less LWP over the great lakes aeras of North America and Europe (Fig. 7b). The increased LWP over the Southern Ocean in the EXP run results in the negative SFCDSW differences (Fig. 6b).

**3.4 Precipitation**

The 120-h averaged non-resolved and resolved precipitation are shown in Fig. 8a and Fig. 8b respectively. The non-resolved precipitation distributes mainly over tropics (Fig. 8a), while the resolved precipitation mainly distributes over the mid-latitude and high-latitude areas (Fig. 8b). This is of course expected since most of the non-resolved precipitation is of convective nature and handled by the GF scheme, which in our tests is not dependent on aerosols. Therefore, the difference of non-resolved precipitation between the EXP and CTL runs is very small and noisy (Fig. 8c). On the other hand, there are significant and interesting differences in the resolved precipitation fields (Fig. 8d). First, the EXP runs have less resolved precipitation over the eastern North America, Europe, and slightly more over the northern oceans. Second, and maybe most interesting, there appears to be significantly more precipitation over the southern high latitudes. The global mean non-resolved and resolved precipitation from the CTL runs are 1.20 mm day$^{-1}$ and 1.90 mm day$^{-1}$ respectively. The global mean non-resolved and resolved precipitation from the EXP runs are very similar with 1.19 mm day$^{-1}$ and 1.99 mm day$^{-1}$ respectively.

When validating against the NOAA CPC rain gauge observation over land, there is a widespread positive precipitation bias over the eastern North America and Europe (Fig. 9a) from the CTL runs, and this positive bias appears improved in the EXP runs (Fig. 9b). The global mean CPC rain gauge observation over land is 1.40 mm day$^{-1}$, the global mean precipitation from the averaged CTL runs over land is 1.78 mm day$^{-1}$ with a 27.1% overestimation, while the global mean precipitation from the averaged EXP runs over land is 1.69 mm day$^{-1}$, reducing the overestimation to 20.7%.

**4 Discussion**

The EXP runs have significantly higher aerosol loading than the CTL runs over Europe, North America and slightly higher aerosol loading over the high latitude Southern Ocean. Interestingly there are opposite resolved precipitation responses to aerosols with less resolved precipitation over North America and Europe and more resolved precipitation over the Southern Ocean. We selected three regions over North America ($95°W - 80°W, 30°N - 45°N$, RegNA), Europe ($25°E - 40°E, 45°N - 60°N$, RegEU), and the Southern Ocean ($90°E - 105°E, 60°S - 45°S$, RegSO) to further analyze the mechanism of aerosol impacts. The total precipitation is almost completely determined by resolved precipitation over these three domains.



Figures 10 a,b,c show some interesting differences and help explain the difference in behavior. There is a significant increase of surface WFA number concentration to about 19 x $10^9$ $kg^{-1}$ over RegNA (Fig. 10a) and RegEU (Fig. 10b), while the surface WFA number concentration is only slightly increased to 2.5 x $10^9$ $kg^{-1}$ over the RegSO (Fig. 10c).

The high aerosol concentration from the EXP run would result in more cloud droplets with reduced diameter, and further reduced the auto conversion and collision coalescence in rain drops and less resolved precipitation over the RegNA and RegEU. The CPC rain gauge observation is 1.31 mm day[-1] over RegEU. When calculating the domain-averaged total precipitation from the CTL runs we get 3.06 mm day[-1], while the average for the EXP runs is 1.34 mm day[-1], significantly less. The CPC rain gauge observations averaged over RegNA show 2.23 mm day[-1]. The domain averaged total precipitation

from the CTL runs is 3.38 mm day[-1] over the same area, and is reduced to 2.12 mm day[-1] when averaged for the EXP runs. It indicates the resolved precipitation response to aerosols over the RegNA and RegEU is from the indirect effect. It looks like the moderate increase of aerosols over the RegSO from the EXP runs enhanced cloud water generation and results in more resolved precipitation. Since there are few CCN over the Southern Ocean, which is away from continental influence, an increase in the number of CCN may significantly impact the microphysics of precipitating clouds (Albrecht 1989). Fan et

al. (2016) also pointed out that recent studies consistently found that adding CCN to warm clouds with very low cloud drop concentrations ($N_d$) can invigorate them to enhance their vertical development, leading to more cloud water content, and enhanced precipitation rates.

## 5. Summary and Conclusions

In this study, a simple and realistic method is used to provide online aerosol emissions for the aerosol-aware TH-E

double moment microphysics in the UFS Weather Model. In TH-E MP, the hygroscopic aerosol is referred to as WFA, and the non-hygroscopic ice-nucleating aerosol is referred to as IFA. We conducted two sets of retrospective runs to examine the indirect aerosol effect through microphysics when using this MP scheme. The CTL experiment applied the GOCART climatology as initial condition for WFA and IFA, and also uses a default empirical formula to compute WFA tendencies. In the runs from the EXP experiment, sea-salt, dust, biomass burning emission modules, and anthropogenic aerosol emissions

are calculated inline without chemical interactions, embedded into the UFS Weather Forecast Model as CCPP-compliant schemes. In this study, the EXP uses initial conditions from the operational GEFS-aerosol model. In operational applications WFA and IFA could be cycled to be independent of GEFS-aerosols. Although there are no additional tracer variables



introduced in this simple and very cost-efficient approach presented here, if a cycled application (no initial conditions from GEFS-aerosol) is preferred, sulfate may be required as one additional tracer variable with some sulfate chemical interactions.

As one possible validation method, we compared AOD forecasts that would result from EXP runs, CTL runs, and AOD analysis from MERRA2 and ECMWF. The 550-nm AOD forecast from the EXP runs are significantly better compared to the CTL run, but further improvements may be necessary to match the MERRA2 and/or ECMWF AOD reanalysis even closer. This may be achieved through improved representation of the emissions as well as wet and dry scavenging. It may also be possible to introduce one or two additional variables to treat wet scavenging differently,

depending on the substance. The closer resemblance of the EXP predictions of AOD to the available analysis products suggests to further test our approach with prognostic aerosol emissions for the aerosol direct effects. The indirect cloud-radiation differences between the EXP and CTL experiments respond in a physically consistent way to the aerosol differences. There are more high, mid, low and total cloud cover fractions from the EXP run over the Northern Hemisphere, and less mid cloud fraction over tropics. In the EXP run, the SFCDSW is stronger over tropics, which corresponds to fewer

aerosols and lower mid cloud fraction. Lower SFCDSW over Northern Hemisphere corresponds to areas with more aerosols and higher cloud fractions in the EXP runs, while lower SFCDSW over the Southern Ocean from the EXP runs appears to be mainly caused by increased cloud water. However, there are distinct differences of resolved precipitation with different aerosol response mechanisms. For the EXP experiment, the very high and significantly increased aerosol number concentrations over Europe and North America result in less resolved precipitation in those areas. On the other hand, there

are very low aerosol number concentrations over the Southern Ocean, with the slight increased aerosols from the EXP runs enhancing cloud water generation to result in more resolved precipitation. Compared to rain gauge observations we were able to significantly improve precipitation biases over Eastern North America and Europe. We also looked at 500 hPa geopotential height Anomaly Correlation Coefficients (ACC), but found only insignificant differences between the two experiments (not shown). This indicates the aerosol difference impact on large-scale circulation is neutral in this study. This

is consistent with the study of implementing a new aerosol climatology dataset into the ECMWF Integrated Forecasting System (IFS) by Bozzo et al. (2020). They reported that the use of a new aerosol climatology has a small impact on the



forecast skill of large-scale weather patterns but has a large local impact on the regional distribution of aerosol radiative forcings.

Computing cost is an important factor preventing coupling of complex chemistry modules with NWP models. Compared with the CTL runs, the additional computing time from the EXP runs is very limited, making this simple and realistic aerosol emission approach very affordable for operational NWP. This study shows that the aerosols in the double-moment microphysics can have significant indirect feedback and impacts on the short-range weather forecasts, and a realistic representation of aerosol emission should be considered in operational NWP. In order to focus on the microphysics response to aerosols, the aerosol-aware feature of GF convection was off in this study, and we will present the convection response to aerosols in a following study. Additionally, we will test the direct radiative impact when using the aerosol optical properties derived from the two double moment microphysics aerosol variables.

**Code and data availability**

The exact version of the UFS Weather Model used to produce the results used in this paper, data for model assessment are available on https://doi.org/10.5281/zenodo.7951581. The aerosol reanalysis of ECMWF-CAMS is from https://www.ecmwf.int/en/research/climate-reanalysis/cams-reanalysis and the aerosol reanalysis of MERRA2 is from https://gmao.gsfc.nasa.gov/reanalysis/MERRA-2/. The CERES radiation flux observation is available on https://ceres.larc.nasa.gov/. The NOAA CPC Daily Precipitation is available on https://psl.noaa.gov/data/gridded/data.cpc.globalprecip.html. The GFS analysis is available on https://www.nco.ncep.noaa.gov/pmb/products/gfs/.

**Financial support**

This research has been supported by the NOAA Global Systems Laboratory (GSL) Base funds, NOAA cooperative agreements NA22OAR4320151.

**Author contributions**

Conceptualization and Methodology: Haiqin Li, Georg Grell, Ravan Ahmadov; Formal analysis and investigation: Haiqin Li, Georg Grell; Funding acquisition and Supervision: Georg Grell; Validation and Visualization: Haiqin Li; Writing –



original draft: Haiqin Li, Georg. A. Grell, Ravan Ahmadov; Writing -review & editing: Li Zhang, Shan Sun, Jordan Schnell, and Ning Wang.

## Competing interests

There are no any competing interests among the authors.

## Disclaimer

The scientific results and conclusions, as well as any views or opinions expressed herein, are those of the author(s) and do not necessarily reflect the views of NOAA or the Department of Commerce.

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



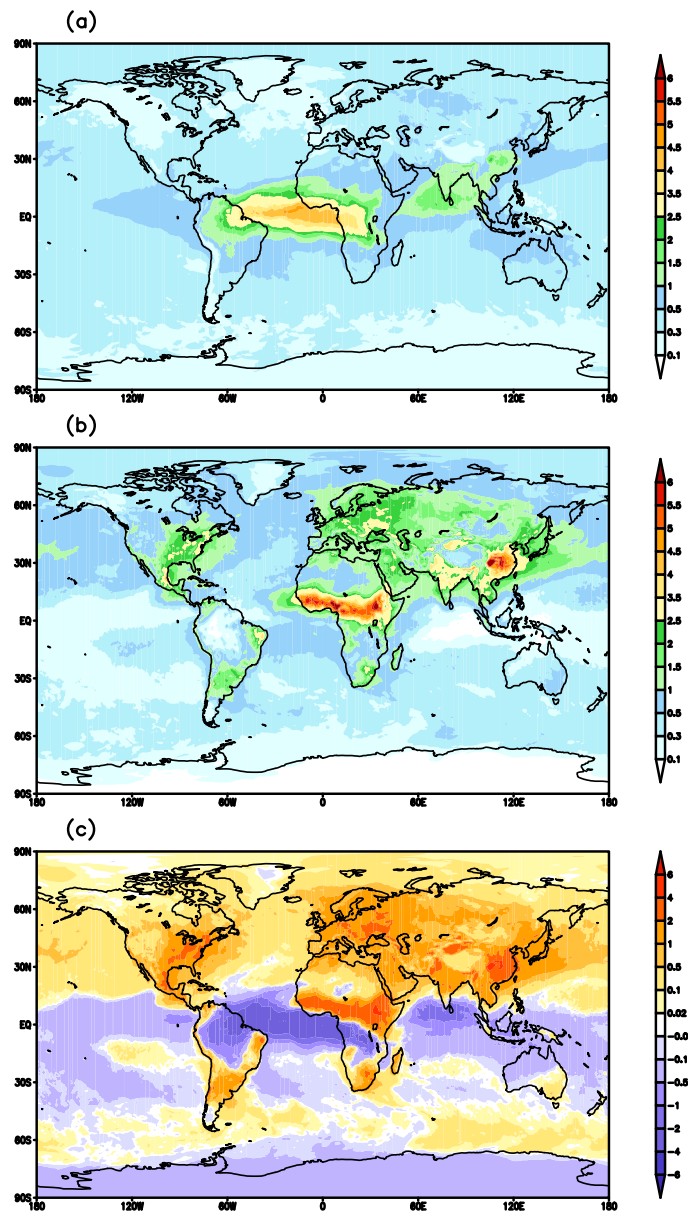

**Figure 1.** The 120-h forecast WFA number concentration ($10^{+13}\,\mathrm{kg}^{-1}$) from (a) CTL, (b) EXP, and (c) EXP minus CTL.






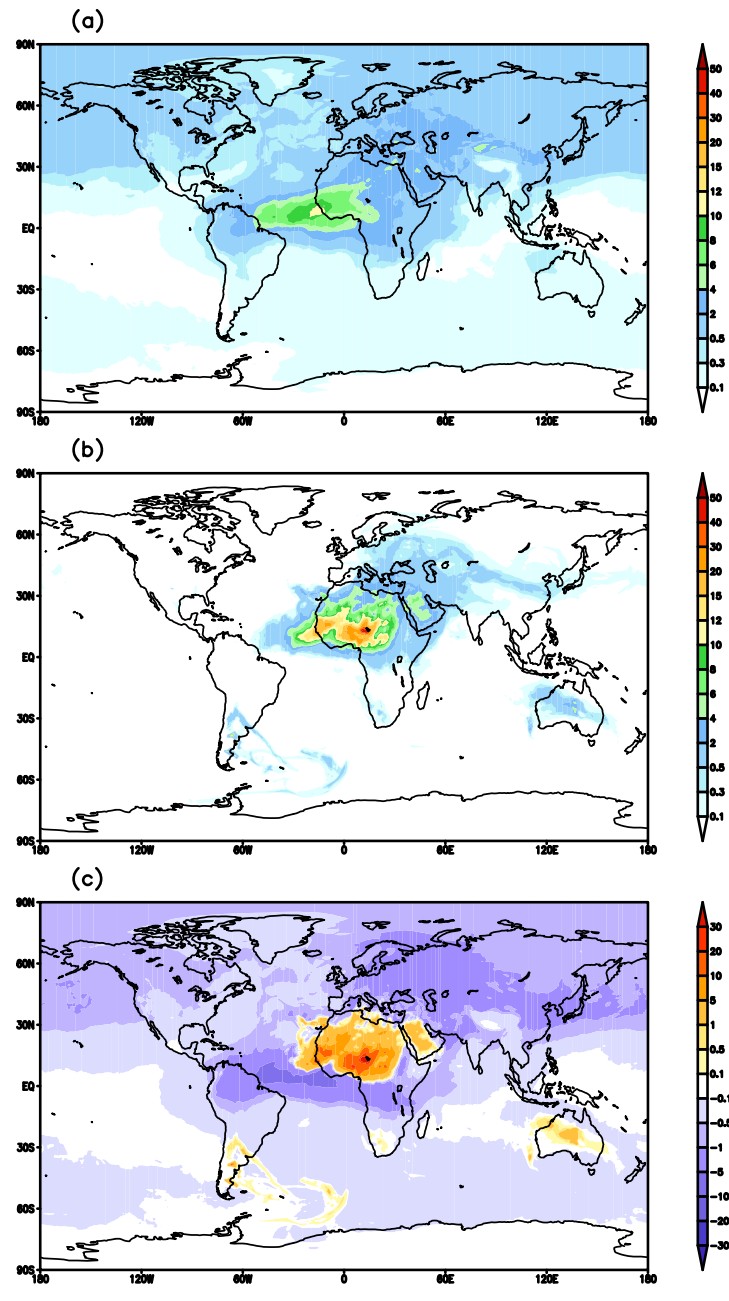

**Figure 2.** The 120-h forecast IFA number concentration ($10^{+10}$ kg$^{-1}$) from (a) CTL, (b) EXP, and (c) EXP minus CTL.




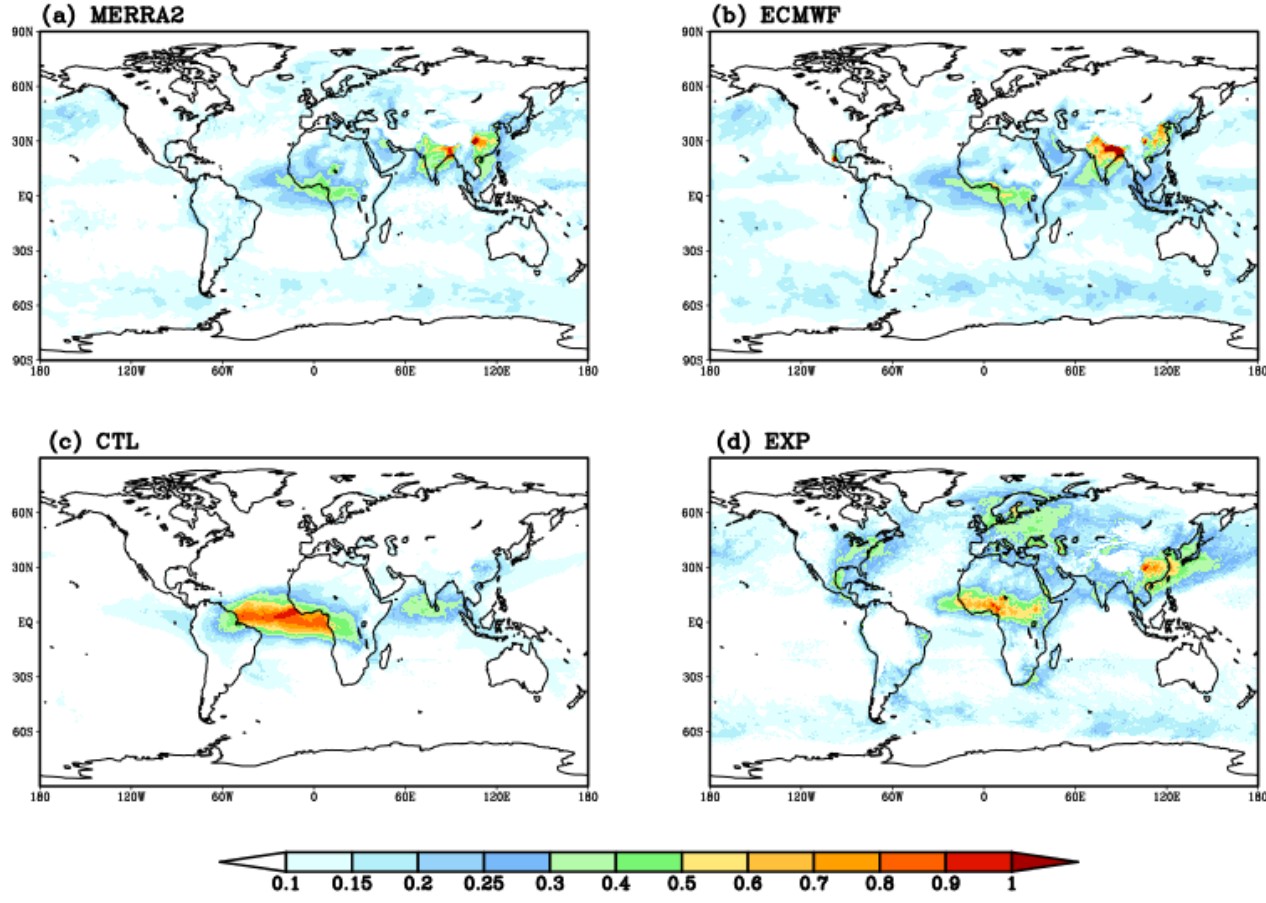

**Figure 3.** The AOD from (a) MERRA2 reanalysis, (b) ECMWF-CAMS reanalysis, and 120-h forecast from (c) CTL, and (d) EXP run.






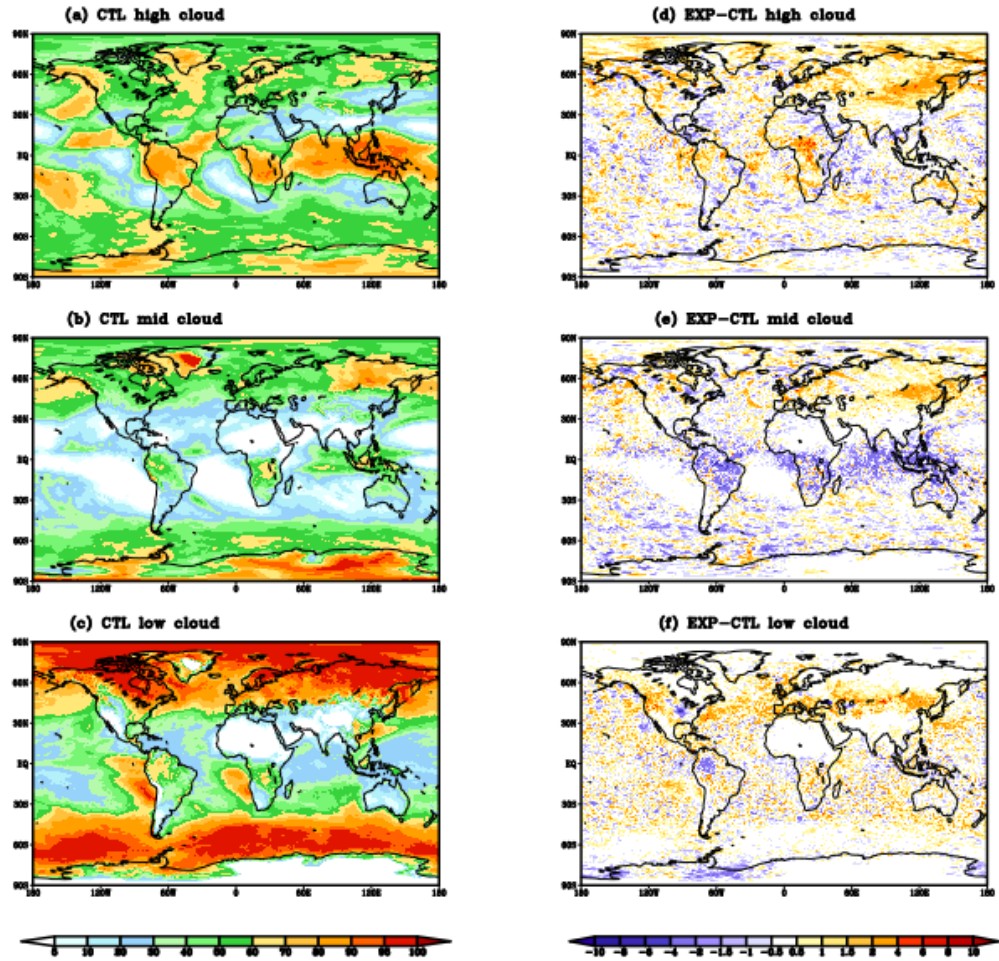

**Figure 4.** The 120-h averaged (a) high cloud, (b) mid cloud, and (c) low cloud fraction (%) from the CTL run, and the differences of (d) high cloud, (e) mid cloud, (f) low cloud fraction (%) between the EXP and CTL runs (EXP minus CTL).





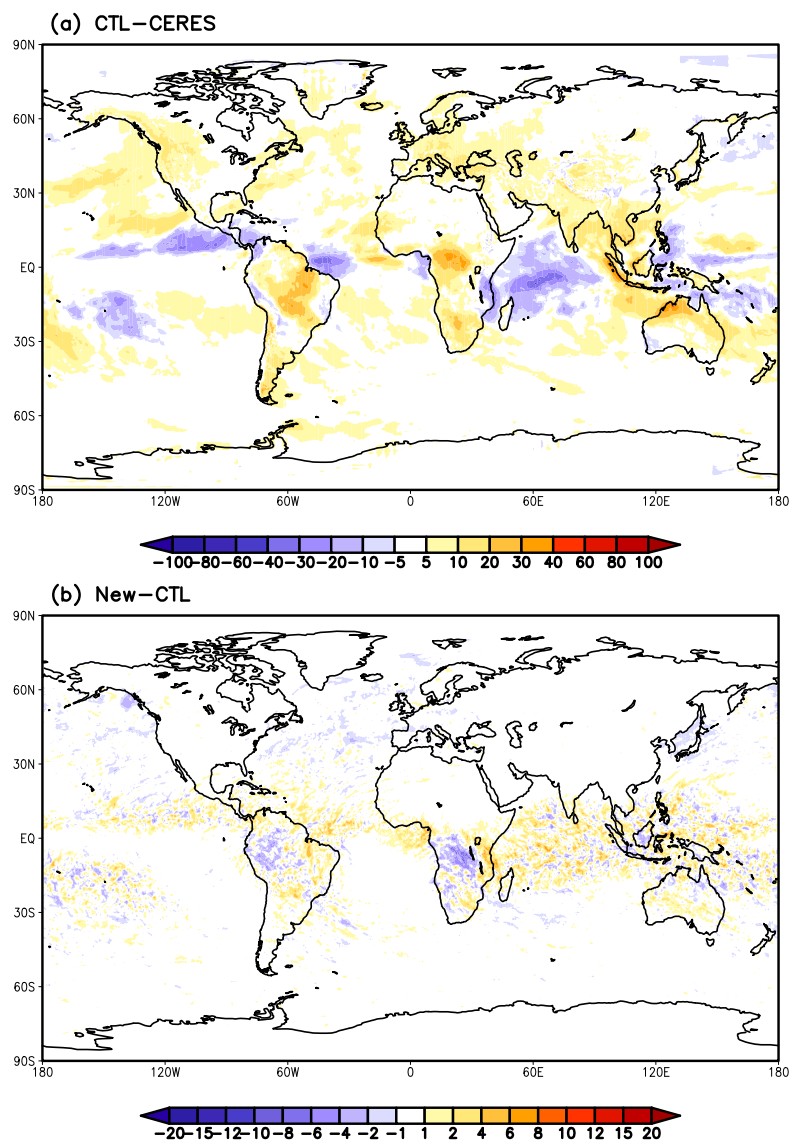

**Figure 5.** The 120-h averaged (a) OLR (W m⁻²) bias from the CTL run against CERES observation, and (b) the difference between the EXP and CTL run (EXP minus CTL).




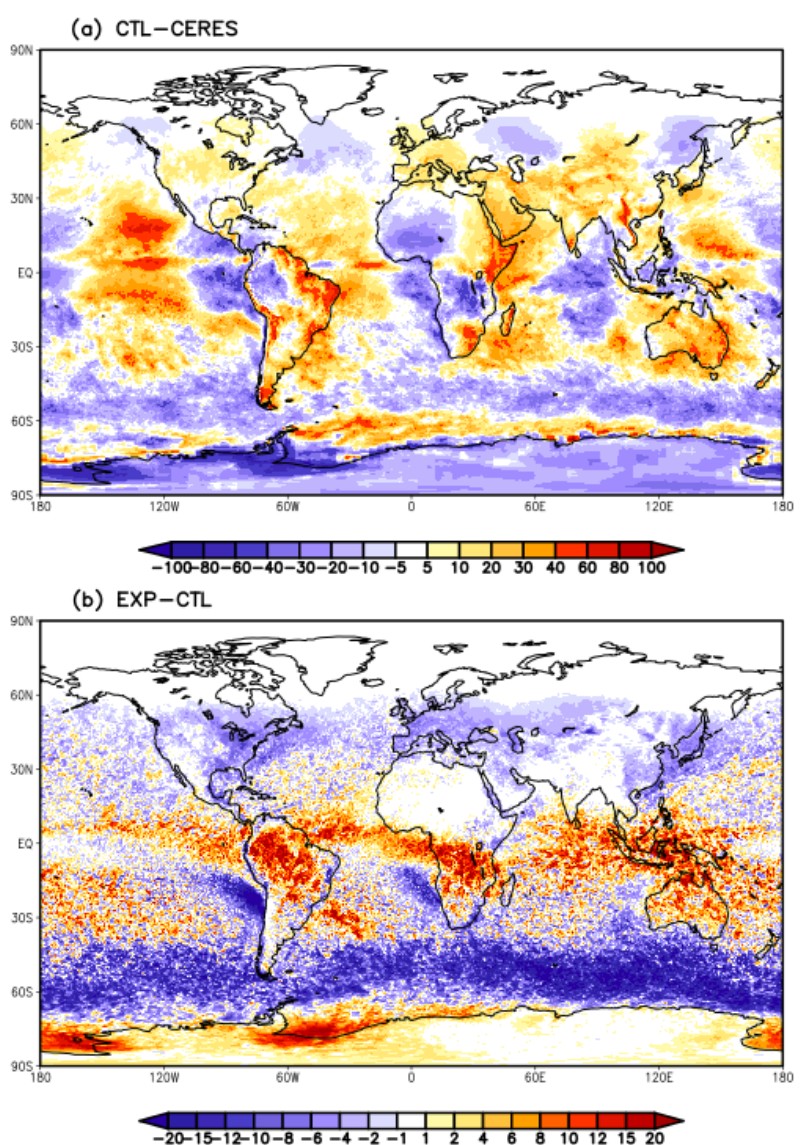

**Figure 6.** The 120h averaged (a) surface downward shortwave radiation (W m$^{-2}$) bias from the CTL run against CERES observation, and (b) the difference between the EXP and CTL run (EXP minus CTL).



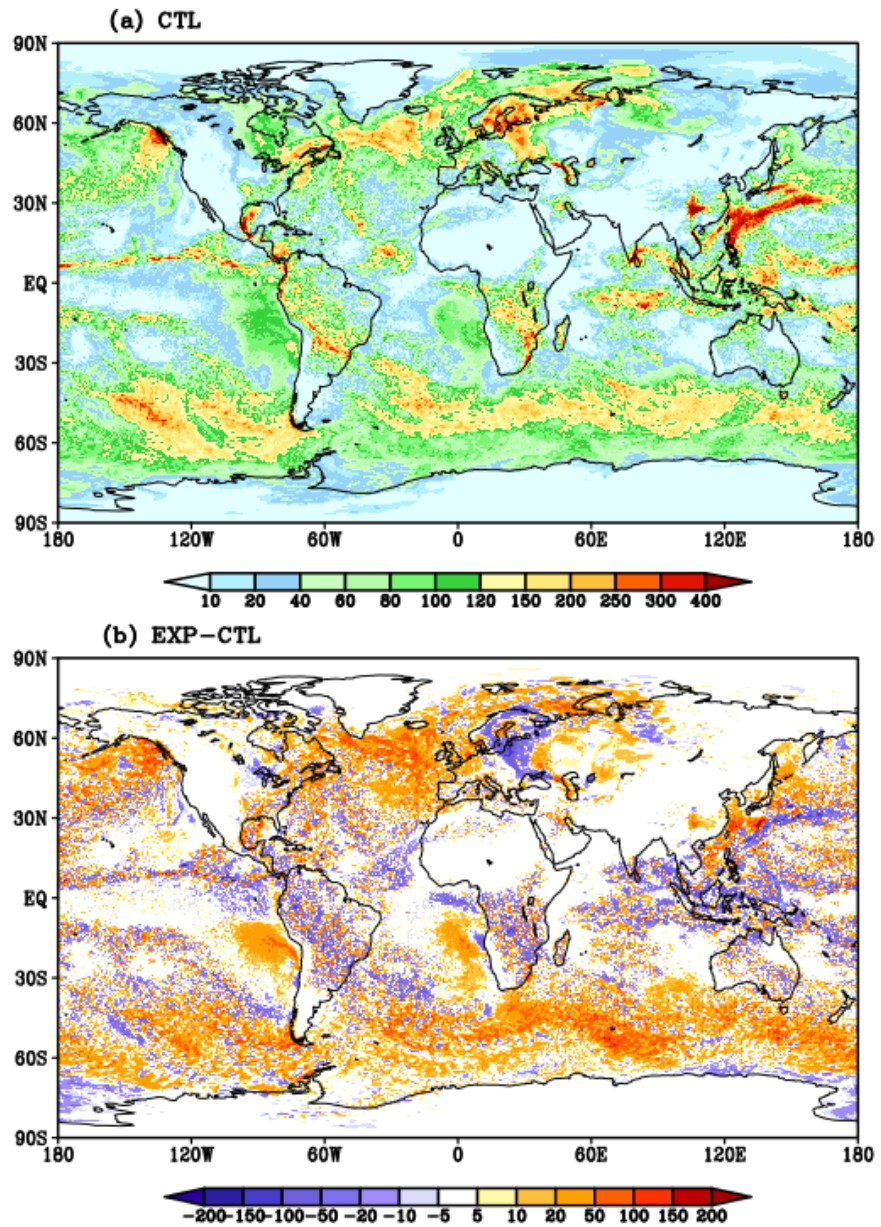

**Figure 7.** The 120-h forecast liquid water path (LWP, g m$^{-2}$) from (a) CTL run, and (b) the difference between EXP and CTL (EXP minus CTL).




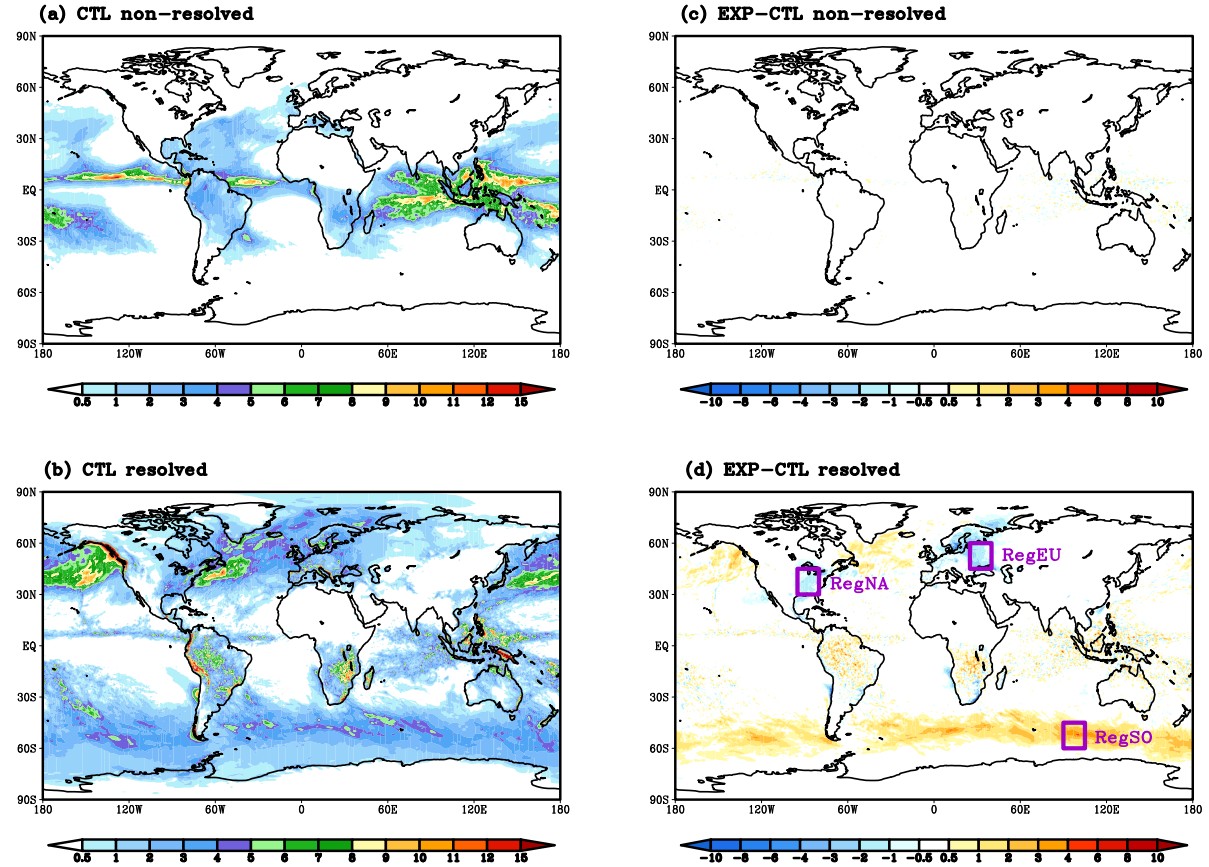

**Figure 8.** The 120-h averaged (a) non-resolved precipitation (mm day$^{-1}$), (b) resolved precipitation (mm day$^{-1}$) from the CTL run, and the differences of (c) non-resolved precipitation (mm day$^{-1}$), (d) resolved precipitation (mm day$^{-1}$) between the EXP and CTL run (EXP minus CTL).

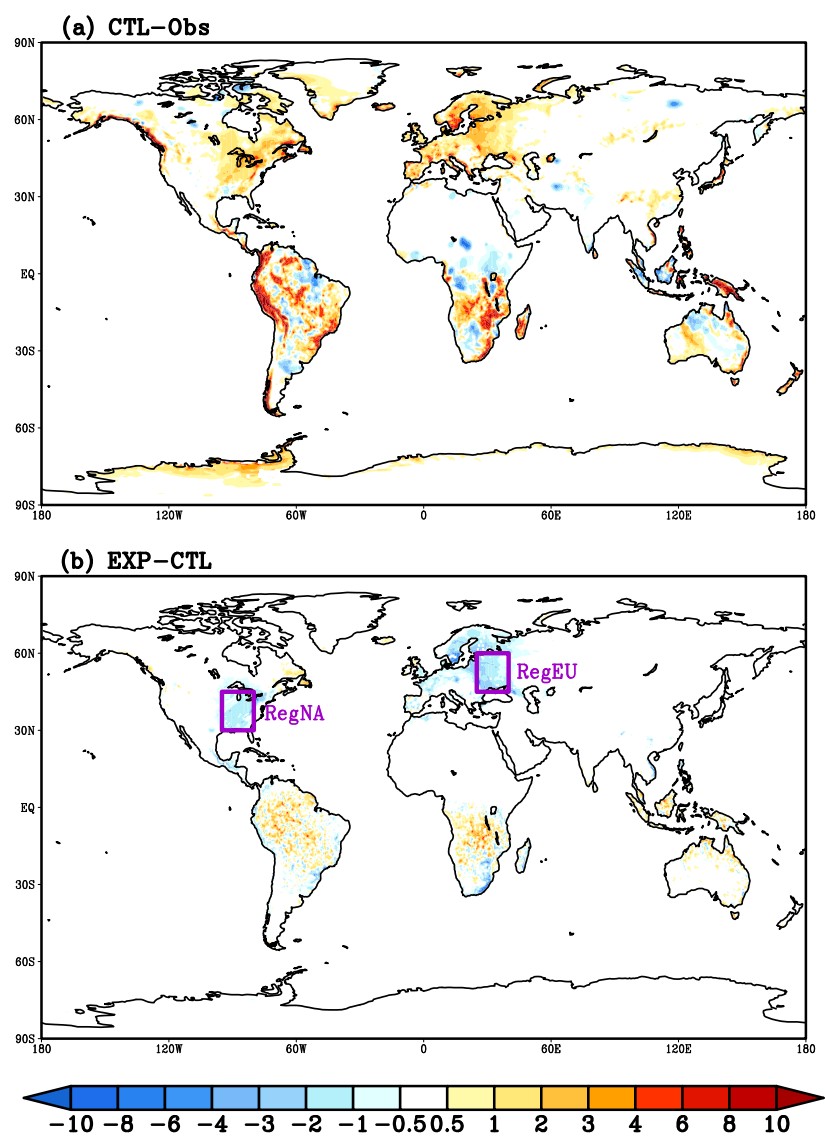

**Figure 9.** The 120-h averaged (a) total precipitation bias (mm day⁻¹) from the CTL run against CPC rain gauge observation, and (b) the difference (mm day⁻¹) between the EXP and CTL run (EXP minus CTL).



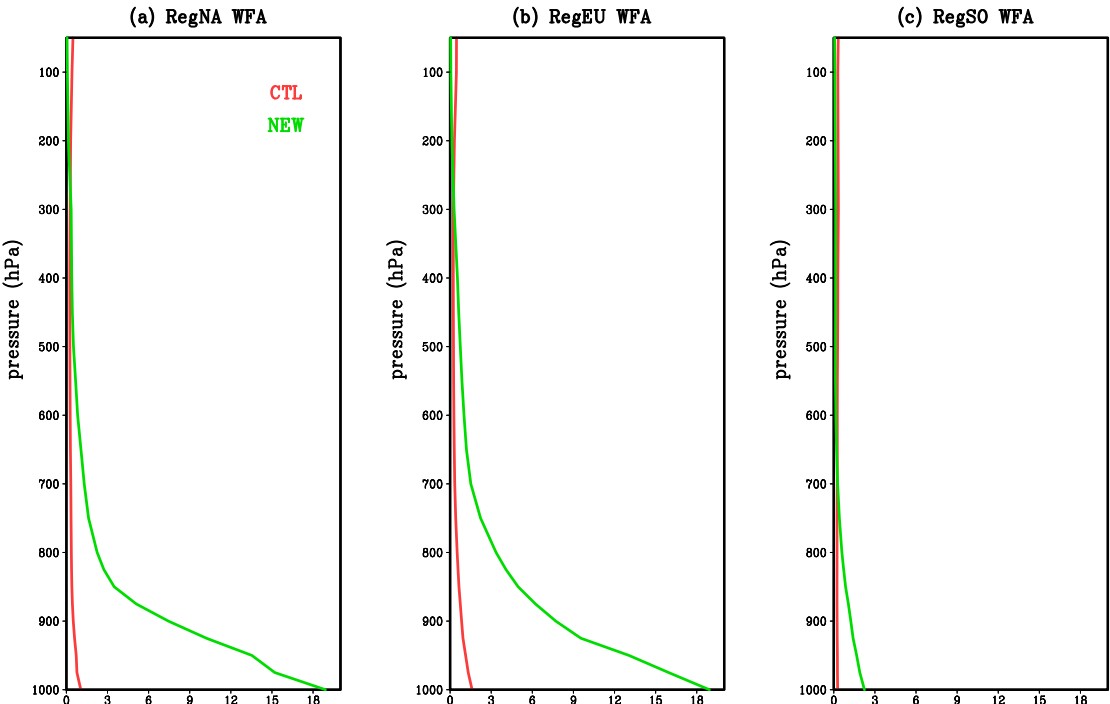

**Figure 10.** The 120-h forecast WFA number concentration ($10^{+9}$ kg$^{-1}$) vertical profile over (a) RegNA, (b) RegEU, (c) RegSO.


**Table 1.** The domain averaged Aerosol Optical Depth (AOD) from 120-h forecast and reanalysis

|  | East Asia (land) (95°E~125°E, 15°N~45°N) | South Asia (land) (65°E~95°E, 5°N~35°N) | Europe (land) (0°E~45°E, 40°N~60°N) | Southern Ocean (0°~360°, 65°S~45°S) |
|---|---|---|---|---|
| **CTL** | 0.11 | 0.10 | 0.08 | 0.06 |
| **EXP** | 0.31 | 0.20 | 0.33 | 0.12 |
| **MERRA2** | 0.26 | 0.30 | 0.12 | 0.12 |
| **ECMWF** | 0.23 | 0.42 | 0.07 | 0.13 |