# Peer review of "A Simple and Realistic Aerosol Emission Approach for use in the Thompson-Eidhammer microphysics scheme in the NOAA UFS Weather Model (version GSL global-24Feb2022)"

_EGUsphere, 2023_

## Referee Comment (RC2)

The author develops a simple and realistic aerosol emission approach. Using the Common Community Physics Package (CCPP), they embedded sea-salt, dust, and biomass burning emission modules as well as anthropogenic aerosol emissions into the Unified Forecast System (UFS) to provide realistic aerosol emissions for these two variables. This approach provides realistic aerosol emissions without the need for additional tracer variables, resulting in minimal additional computing cost.

The manuscript is well-written and easy to follow. The new aerosol emission approach demonstrates good realism in the runs with online emissions, as evidenced by comparisons with analyzed fields for the Aerosol Optical Depth (AOD). Additionally, the cloud cover, radiation, and precipitation in the runs also exhibit realistic representations. This simple and realistic aerosol emission approach is highly suitable for operational Numerical Weather Prediction (NWP) due to its affordability.

The reviewer recommends a minor revision for the manuscript. Below are the main comments of the reviewer.

**Specific comments:**

1. Please ensure consistency in the citation format of the references mentioned in the paper. For example, Line 36: "Conrick et al. 2021" should be "Conrick et al. (2021)"; Line 44: "Zhao et al. 2021" should be "Zhao et al. (2021)"; Line 49: "Mulcahy et al. 2014" should be "Mulcahy et al. (2014)" .

2. In the main text, the full name should be provided for the first occurrence of an abbreviation. Please check this. For example, "UFS" should be expanded to its full name at Line 61, rather than at Line 69.

3. Will the UFS Weather Model (https://github.com/NOAAGSL/ufs-weather-model/releases/tag/global-24Feb2022) in this study be made publicly available? Currently, it seems that the provided link is not accessible.

4. Please provide a detailed explanation of the $fact_{wra\_ss}$、$fact_{wra\_oc}$、$fact_{ifa}$ in the formula section.

5. The author should provide an accurate description of the experimental results. e.g., in Figure 3, the EXP experiment overestimated AOD in central Africa while underestimating it in South Asia.

6. To provide a more intuitive comparison between CTL and EXP, it would be helpful to include the EXP minus CERES results in Figure 5 and Figure 6. Additionally, in Figure 5b, the title "New-CTL" should be changed to "EXP-CTL" to maintain consistency with the other figures. Similarly, in Figure 10a, "NEW" should be changed to "EXP" for consistency.

7. Just like in Figures 5 and 6, it would be beneficial to include the results of EXP minus Obs in Figure 9. Comparing CTL minus Obs and EXP minus Obs would provide a more comprehensive assessment of the improvement and differences between EXP and CTL.

8. Lines 152-154: Is there any difference in the calculation of the averages for WFA, IFA, AOD, temperature, hydrometeors, cloud cover, radiation, and precipitation, are they all averaged over a 120-hour forecast period? Does "The forecast is integrated for 120 h" mean that the forecast has a time resolution of 120 hours?

9. Lines 192-194: "The global mean SFCDSW from the averaged CERES observations compared to the CTL and EXP runs are 192.5 W m$^{-2}$ , 194.7 W m$^{-2}$ and 193.7 W m$^{-2}$, respectively" means the

global mean SFCDSW from the averaged CERES observations, the CTL, the EXP are 192.5 W $m^{-2}$ , 194.7 W $m^{-2}$ and 193.7 W $m^{-2}$, respectively? What does "comparison" mean here? in the same way What does "compare" mean in 199-201? Please provide a clear description.

10. Line 194 and Line 200, "W m$^{-2}$" should be "W $m^{-2}$."

11. The validity of the liquid water path (LWP) obtained from the CTL and EXP runs can be assessed by comparing it with other LWP products or by calculating LWP using ERA-Interim/CFSR (Climate Forecast System Reanalysis) data. Additionally, the results of the CTL and EXP runs for the 120-h averaged high cloud, mid cloud, and low cloud cover fractions can also be compared with satellite products such as CloudSat/CALIPSO.

12. Some figures are not very clear, such as Figure 4. It is recommended to replace them with clearer versions for better readability by the readers.

---

## Author Response (AR1)

Referee #1
**We appreciate the reviewer's careful reading of the manuscript and the constructive comments. We revised the manuscript following the reviewer's suggestions.**

This paper introduces a change to allow the microphysics and radiation in the UFS Weather Model to interact with cloud and ice nuclei from real-time emission sources rather than the climatology currently used with the Thompson scheme. The work represents a step forwards in allowing basic aerosol-physics interactions without the use of a full chemistry model for medium-range global applications. A series of cases are tested and compared with the climatology option and verifying data from various sources. The results show that this is a promising direction in reducing some biases and adding more realism to aerosol effects globally. The paper is well presented and worth publication. I only have some minor comments to address.

Minor Points
1. line 105. State the units of WFA and IFA. Equations are hard to interpret without units.
→ **Following the reviewer's suggestion, the units of WFA and IFA have been added in the revised manuscript "The tendencies of WFA (kg$^{-1}$) and IFA (kg$^{-1}$)".**

2. L114-116. 1^-9 is a typo.
→ **Thanks to the reviewer for pointing this out, and it has been corrected to "10^-9".**

3. L123. "is a scale-aware"
→ **Thanks to the reviewer for pointing this out, and it has been corrected to "is scale-aware".**

4. L127. "boulder"
→ **Thanks to the reviewer for pointing this out, and it has been corrected to "boundary" in the revised manuscript.**

5. L127. Does MYNN PBL mix WFA and IFA?
→ **Yes, it has been clarified in the revised manuscript "to represent PBL mixing (including WFA and IFA)".**

6. L131. "surface climatological(?) number concentration"
→ **Yes, it has been clarified as "surface climatological number concentrations".**

7. Figure 1. Vertically integrated?
→ **Yes, it is clarified as "vertically integrated" for the caption of Figure 1 in the revised manuscript.**

8. L169. Is AOD interactive with radiation in the model?
→ **We focused on the aerosol indirect feedback to microphysics, and the AOD derived from WFA and IFA is not interactive with radiation in the model configuration used in this study. We clarified this in the revised manuscript as "Overall, the AOD derived from WFA and IFA is in reasonable**

**agreement with analysis, suggesting it may be worth investigating the direct radiation impact in comparison to using climatologies in the future studies.".**

9. L186-189. "aeras" typo.
→ **Thanks to the reviewer for pointing this out, and it has been corrected as "areas".**

10. L233. Not sure how to relate 10^9 profiles with 10^13 integrated values?
→ **The profiles of WFA in Figure 10 are in unit of number per kg, and the vertical integral of WFA (number per m^2 ) is by the sum of WFA*delp/g in each vertical layer. delp is the layer pressure in the unit of Pa, and g is gravitational acceleration. Thus, we get the 10^13 vertically integrated values in Figure 1.**

11. L246. Invigoration related to freezing level in cloud?
→ **Thanks to the reviewer, we added the following**
**text in the revised manuscript "Increased cloud water may also lead to increased freezing and latent heating above the freezing level, further invigorating the clouds".**

12. L251. Do non-hygroscopic aerosols include black carbon which interacts with radiation but is not an ice nucleus?
→ **Non-hygroscopic aerosol (ice-friendly aerosol) is primarily considered to be dust (Thompson and Eidhammer 2014).**

Referee # 2

**We appreciate the reviewer's careful reading of the manuscript and the constructive comments. We've revised the manuscript following the reviewer's suggestions.**

The author develops a simple and realistic aerosol emission approach. Using the CommonCommunity Physics Package (CCPP), they embedded sea-salt, dust, and biomass burning emission modules as well as anthropogenic aerosol emissions into the Unified Forecast System(UFS) to provide realistic aerosol emissions for these two variables. This approach provides realistic aerosol emissions without the need for additional tracer variables, resulting in minimal additional computing cost.

The manuscript is well-written and easy to follow. The new aerosol emission approach demonstrates good realism in the runs with online emissions, as evidenced by comparisons with analyzed fields for the Aerosol Optical Depth (AOD). Additionally, the cloud cover, radiation, and precipitation in the runs also exhibit realistic representations. This simple and realistic aerosol emission approach is highly suitable for operational Numerical Weather Prediction (NWP) due to its affordability.

The reviewer recommends a minor revision for the manuscript. Below are the main comments of the reviewer.

Specific comments:

1. Please ensure consistency in the citation format of the references mentioned in the paper. For example, Line 36: "Conrick et al. 2021" should be "Conrick et al. (2021)"; Line 44: "Zhao et al. 2021" should be "Zhao et al. (2021)"; Line 49: "Mulcahy et al. 2014" should be "Mulcahy et al. (2014)" .
→ **Thanks to the reviewer for pointing this out. The references have been reformatted in the revised manuscript.**

2. In the main text, the full name should be provided for the first occurrence of an abbreviation. Please check this. For example, "UFS" should be expanded to its full name at Line 61, rather than at Line 69.
→ **Thanks to the reviewer for pointing this out. The abbreviations have been rearranged in the revised manuscript.**

3. Will the UFS Weather Model (https://github.com/NOAAGSL/ufs-weather-model/releases/tag/global-24Feb2022) in this study be made publicly available? Currently, it seems that the provided link is not accessible.
→ **I am sorry that https://github.com/NOAA-GSL/ufs-weather-model/releases/tag/global-24Feb2022 under the github organization of NOAA-GSL is not accessible publicly. The exact version of the UFS Weather Model used to produce the results used in this paper is available on https://doi.org/10.5281/zenodo.7951581.**

4. Please provide a detailed explanation of the factwra_ss、 factwra_oc、 factifa in the formula section.

→ **Following the reviewer's suggestion, a detailed explanation has been added in the revised manuscript that "The $fact_{wfa\_ss}$, $fact_{wfa\_oc}$, and $fact_{ifa}$ are the tuning factors with the diameters of sea salt ($wfa\_diameter\_ss$), organic carbon ($fact_{wfa\_oc}$) and dust ($fact_{ifa}$), respectively".**

5. The author should provide an accurate description of the experimental results. e.g., in Figure 3, the EXP experiment overestimated AOD in central Africa while underestimating it in South Asia.

→ **Following the reviewer's suggestion, we added the description of AOD in the revised manuscript that "The AOD from the EXP run is somewhat overestimated over Eastern Europe, the Eastern part of the US, and central Africa, with some underestimation over South Asia (Table 1)".**

6. To provide a more intuitive comparison between CTL and EXP, it would be helpful to include the EXP minus CERES results in Figure 5 and Figure 6. Additionally, in Figure 5b, the title "New-CTL" should be changed to "EXP-CTL" to maintain consistency with the other figures. Similarly, in Figure 10a, "NEW" should be changed to "EXP" for consistency.

→ **Following the reviewer's suggestion, the EXP minus CERES results are added in Figure 5 and Figure 6. The "NEW" has also been corrected to "EXP" for consistency in Figure 5b and Figure 10a.**

7. Just like in Figures 5 and 6, it would be beneficial to include the results of EXP minus Obs inFigure 9. Comparing CTL minus Obs and EXP minus Obs would provide a more comprehensive assessment of the improvement and differences between EXP and CTL.

→ **Following the reviewer's suggestion, the EXP minus Obs has been added in Figure 9.**

8. Lines 152-154: Is there any difference in the calculation of the averages for WFA, IFA, AOD, temperature, hydrometeors, cloud cover, radiation, and precipitation, are they all averaged over a 120-hour forecast period? Does "The forecast is integrated for 120 h" mean that the forecast has a time resolution of 120 hours?

→ **We clarified in the revised manuscript that "The WFA, IFA, AOD, temperature, and hydrometeors are from the instantaneously values at 120 h forecast. The cloud cover, radiation, and precipitation are from the cumulative values over the 120-h forecast period"**

9. Lines 192-194: "The global mean SFCDSW from the averaged CERES observations compared to the CTL and EXP runs are 192.5 W m-2 , 194.7 W m-2 and 193.7 W m-2 , respectively" means the global mean SFCDSW from the averaged CERES observations, the CTL, the EXP are 192.5 Wm-2 , 194.7 W m-2 and 193.7 W m-2 , respectively? What does "comparison" mean here? In the same way What does "compare" mean in 199-201? Please provide a clear description.

→ **Thanks to the reviewer for pointing this out, it has been clarified in the revised manuscript as " The global mean SFCDSW from the averaged CERES observations , the CTL, and the EXP runs are 192.5 W m$^{-2}$, 194.7 W m$^{-2}$ and 193.7 W m$^{-2}$ respectively".**

10. Line 194 and Line 200, "W m_2" should be "W m-2 ."

→ **Thanks to the reviewer for pointing this out, the "W m_2" has been corrected to " W m$^{-2}$ " in Line 194 and Line 200 of the revised manuscript.**

11. The validity of the liquid water path (LWP) obtained from the CTL and EXP runs can be assessed by comparing it with other LWP products or by calculating LWP usingERA-Interim/CFSR (Climate Forecast System Reanalysis) data. Additionally, the results of the CTL and EXP runs for the 120-h averaged high cloud, mid cloud, and low cloud cover fractions can also be compared with satellite products such as CloudSat/CALIPSO.

→ **Following the reviewer's suggestion, the LWP from ERA5 and GFS Analysis are added in Figure 7, and the CERES satellite products of high cloud, mid cloud and low cloud cover are added in Figure 4 of the revised manuscript.**

12. Some figures are not very clear, such as Figure 4. It is recommended to replace them with clearer versions for better readability by the readers.

→ **Following the reviewer's suggestion, all the figures are replaced with PDF formats for better readability.**